# The relationship between participation in leisure activities and incidence of falls in residential aged care

**Guogui Huang**[1]*, **Nasir Wabe**[1], **Magdalena Z. Raban**[1], **S. Sandun Malpriya Silva**[1], **Karla Seaman**[1], **Amy D. Nguyen**[1,2], **Isabelle Meulenbroeks**[1], **Johanna I. Westbrook**[1]

**1** Centre for Health Systems and Safety Research, Australian Institute of Health Innovation, Macquarie University, Sydney, New South Wales, Australia, **2** St Vincent's Clinical School, UNSW Medicine, UNSW Sydney, Sydney, New South Wales, Australia

* guogui.huang@mq.edu.au

**Data Availability Statement:** The data used in this study were collected by the aged care provider and restrictions apply to the availability of these data. As part of the approval issued by Macquarie

## Abstract

### Background

Active engagement in leisure activities has positive effects on individuals' health outcomes and social functioning; however, there is limited understanding of the link between participation in leisure activities, particularly non-exercise activities, and falls in older adults. This study aimed to determine the relationship between participation in leisure activities and the incidence of falls, and the variation of this relationship by dementia status in residential aged care facilities (RACFs).

### Methods

A retrospective longitudinal cohort study utilising routinely collected data (January 2021-August 2022) from 25 RACFs in Sydney, Australia, was conducted. The cohort included 3,024 older permanent residents (1,493 with dementia and 1,531 without) aged ≥65 and with a stay of ≥1 week. The level of participation in leisure activities was measured using the number of leisure activities per 1,000 resident days and divided into quartiles. Outcome measures were the incidence rate of all falls and injurious falls (i.e., number of falls per 1,000 resident days). We used multilevel negative binary regression to examine the relationship between leisure participation and fall incidence.

### Results

For the whole sample, leisure participation was significantly inversely associated with the incidence rate of all falls and injurious falls. For example, residents in the high leisure participation group were 26% less likely to experience a fall compared to those in the low leisure participation group after controlling for confounders (incidence rate ratio = 0.74, 95% confidence interval = 0.60, 0.91). Such inverse relationship was observed in both exercise and non-exercise activities and was stronger among residents without dementia.

University Human Research Ethics Committee
Medical Sciences Committee (Ref No:
52019614412614), 'All research data (e.g.
interviews, data extractions) will be stored
electronically on Macquarie University's secure
password-protected server' and 'Only the named
project team members approved on the ethics
application will have access to the data extracted
and collected during the period of the research
project', given that data contain potentially
identifying or sensitive patient information.
Therefore, the data used in this study cannot be
deposited in a public data repository. Data are
however available from the authors (Guogui
Huang, guogui.huang@mq.edu.au) or through the
Centre of Health Systems and Safety Research,
Macquarie University (chssr@mq.edu.au) upon
reasonable request and with permission of aged
care provider.

**Funding:** This work was supported by the
Australian National Health and Medical Research
Council (NHMRC) Partnership Project Grant
(1170898). The funding organization did not have
any influence on the study design, data collection,
analysis and interpretation as well as the
preparation, review, or approval of the manuscript
for publication.

**Competing interests:** The authors have declared
that no competing interests exist.

## Conclusions

Leisure participation is associated with a lower rate of falls, a key quality indicator by which
RACFs are benchmarked and funded in Australia and many other countries. More recognition and attention are needed for the currently underfunded leisure activities in RACFs in
future funding arrangement.

## Introduction

Leisure activities are defined as activities that people engage in during their free time for the
purposes of relaxation, entertainment or personal growth [1, 2]. Engagement in leisure activities plays a key role among older adults given its strong effects on individuals' health outcomes
and social functioning [3, 4]. Benefits of leisure activities for older adults include decreased
depression symptoms and loneliness [5, 6], lower mortality [7], slowed decline in cognitive
and physical function [8–10], improved subjective wellbeing [11] and close relationships with
family and friends [12, 13]. Thus, active engagement in personally meaningful leisure activities
has long been considered an important component of successful aging [12, 14].

Falls are defined as "inadvertently coming to rest on the ground, floor or other lower level,
excluding intentional change in position to rest in furniture, wall or other objects" [15]. Falls
among older adults are a prominent public health problem globally, with >25% of older community-dwellers [15] and >50% of residential aged care facility (RACF) (i.e., nursing homes
and long-term care facilities) residents [16] falling annually. Falls can cause severe medical,
psychological and financial consequences for older adults [15, 17–19].

Despite ample evidence of the beneficial effects of leisure activities on participants' health
and wellbeing, understanding of whether and how leisure participation affects falls among
older adults has been inconclusive [20, 21]. Previous studies demonstrate that leisure participation can both positively and negatively affect fall outcomes. Specifically, leisure participation
can improve strength, balance, bone mineral density and coordination, and hence, contribute
to a decreased fall risk. For example, a four-year randomized controlled trial in the United
States (US) reported a reduced fall risk among people aged $\geq$65 (in both community and
RACF settings) after participating in a physical exercise program for 10–36 weeks compared
to those who did not participate (incidence rate ratio [IRR] = 0.90, 95% confidence interval
[CI] = 0.81, 0.99) [22]. Similarly, a randomized controlled trial in New Zealand reported a
reduced fall risk among women aged $\geq$80 after enrolling in a one-year leisure physical exercise
program at home compared to those similarly aged but not enrolled (hazard ratio = 0.68, 95%
CI = 0.52, 0.90) [23]. However, leisure participation also exposes older adults to some fall risk
factors, such as fatigue and risky environments (e.g., stairs), and thus, might offset the positive
effects on fall risk [20]. For example, one study based on a randomized controlled trial in the
community setting in Thailand reported a reduced fall risk among people aged $\geq$65 enrolling
in a one-year physical exercise program compared to those not enrolled; however, the confidence intervals were wide and therefore not significant (hazard ratio = 0.84, 95% CI = 0.58,
1.22) [24]. There are also a few studies (in cohort design with a follow-up period from 0.5 to 5
years) that indicate a U-shaped relationship between leisure participation and fall risk among
older adults, reporting only medium-level leisure participation is linked to a decreased fall risk
[25, 26].

However, when exploring the relationship between leisure participation and falls among
older adults, these existing studies tended to focus on exercise activities. What has rarely been

investigated is how non-exercise leisure activities (e.g., playing card games) are associated with fall outcomes among older adults. It is of critical importance to include non-exercise leisure activities in studies on the relationship between leisure activities and falls, given that non-exercise leisure activities are common in later life and that non-exercise leisure activities importantly affect individuals' physical and psychological outcomes and quality of life [11, 27–30]. The overall positive effect of non-exercise leisure participation on individuals' physical and psychological outcomes, well documented in the existing literature [11, 27–30], might subsequently positively affect fall outcomes. Specifically, engagement in both exercise and non-exercise leisure activities (e.g., reading, playing board games and playing musical instruments) were found to be significantly associated with a reduced risk of dementia [30], which might therefore affect susceptibility to falls. For example, a recent systematic review that covers 38 longitudinal studies, with 2,154,818 participants at baseline, indicated that physical exercises (e.g., walking and jogging) (rate ratio = 0.83, 95% CI 0.78–0.88), cognitive activities (e.g., reading, playing games and playing music instruments) (rate ratio = 0.77; 95% CI 0.68–0.87) and social activities (e.g., participating in volunteer work and meeting relatives or friends) (rate ratio = 0.93; 95% CI 0.87–0.99) were all associated with a decreased incidence of all-cause dementia [31]. Dementia is a strong predictor of falls due to a greater risk of people with dementia experiencing problems with mobility and balance and difficulties with their memory, along with side effect of medication prescribed for dementia (e.g., dizziness and sedation), behavioural and psychological symptoms associated with dementia (e.g., agitation) and challenges for individuals with dementia to navigate complex or unfamiliar environments [32, 33]. The inverse relationship between participation in exercise and non-exercise leisure activities and dementia implies a potential positive impact of these activities on falls. Additionally, the probability of engaging in non-exercise leisure activities increases with age among older adults. Previous evidence demonstrated a steep decrease in participation in physical leisure activities (e.g., exercise sports) after age 65, but a steady increase in participation in cognitive/sedentary leisure activities (e.g., reading and playing puzzles/chess) between ages 50–75 [34]. Therefore, it is critical to include non-exercise activities to fully understand the relationship between leisure participation and falls among older adults. This is particularly important for the setting of RACFs, where residents are generally in very old age and few studies have been conducted to examine the effect of leisure participation on falls in this setting [23–26].

To fill this gap, we aimed to investigate the association of exercise and non-exercise leisure participation on the rate of falls in older adults living in RACFs. Leisure activities are an important part of life of RACF residents and are often delivered in groups. These activities include a range of types, such as community outings, walking/exercise groups and social games (e.g., trivia, bingo and cards). Our specific objectives were to examine the relationship between such leisure participation and falls and the variations of this relationship by dementia status. The knowledge produced could provide new insights into how falls prevention and intervention programs in RACFs could be strengthened, and inform resourcing of leisure activities for RACF residents.

## Methodology

### Study design

We conducted a retrospective longitudinal cohort study using routinely collected aged care data obtained from 25 RACFs of a large not-for-profit aged care provider in New South Wales, Australia. The study period was from 1 January 2021 to 30 August 2022. During this period, a resident might have multiple records of admission since individuals could enter, leave and re-

enter the facilities at any time. The study was reviewed and approved by Macquarie University Human Research Ethics Committee (Application No. 520221251343996).

## Sample

The extracted data contained de-identified information of 3,946 admissions from 3,782 residents aged ≥65 years. Because de-identified data were used, requirement of obtaining participants' consent was waived by Macquarie University Human Research Ethics Committee. We excluded data about residents who only used interim care (n = 2), individuals who were discharged on the same date of admission (n = 59), respite residents (n = 663) and those whose length of stay was shorter than one week (n = 38). This yielded a sample of 3,024 residents (1,493 with dementia and 1,531 without dementia) with 3,128 admissions. The sample selection process is presented in Fig 1.

## Data sources

We utilized routinely collected aged care data containing resident profile data, fall incident reports and information on leisure activities. Routinely-collected data are those collected electronically through the information system of the aged care provider on a day-to-day basis to support the delivery of care services [35]. The profile dataset provided information of residents' basic demographic characteristics (e.g., year of birth and gender), health-related information (e.g., whether having diabetes and hypertension) and admission records (e.g., resident ID, facility ID and date of admission/departure). The fall incident data comprized all fall events occurring in facilities, including incident date, location (e.g., resident's room and bathroom/toilet) and consequences (e.g., whether causing injuries). The leisure activities data contained information on leisure activities that RACFs organized for residents, including the date, attendance, type of leisure activities and residents' participation level. Records of 'unattended' or 'attended but refused to participate' were removed when we counted the number of leisure activities. These datasets have been utilized in previous publications [36, 37].

## Outcome measures

The outcome measures were the incidence rate of all falls and injurious falls. All falls were defined as any fall in the incident dataset regardless of the fall consequences (i.e., causing injuries or needing hospitalization). Injurious falls were falls that caused any type of injury (e.g., head injury, hip injury or deep issue injury).

## Measures of leisure activities

Leisure activities were categorized into exercise activities and non-exercise activities. Exercise activities include physical exercise/sports activities, such as walking, dancing, Tai-chi, yoga and ball sports, while non-exercise activities comprise leisure activities for entertainment/socialising/religious purposes, such as floor games, table games, coffee chat and bible study. For details of the categorization please refer to Table 1.

## Statistics analysis

We calculated incidence rates (i.e., per 1,000 resident days) for both outcome measures and leisure activities. Incidence rate for each RACF resident was based on their number of incidents (i.e., falls or leisure activities) occurring during their stay. For the small number of residents with multiple admissions (n = 149), their incidence rate was calculated taking into account all the incidents occurring in their multiple admissions and the total length of stay of all

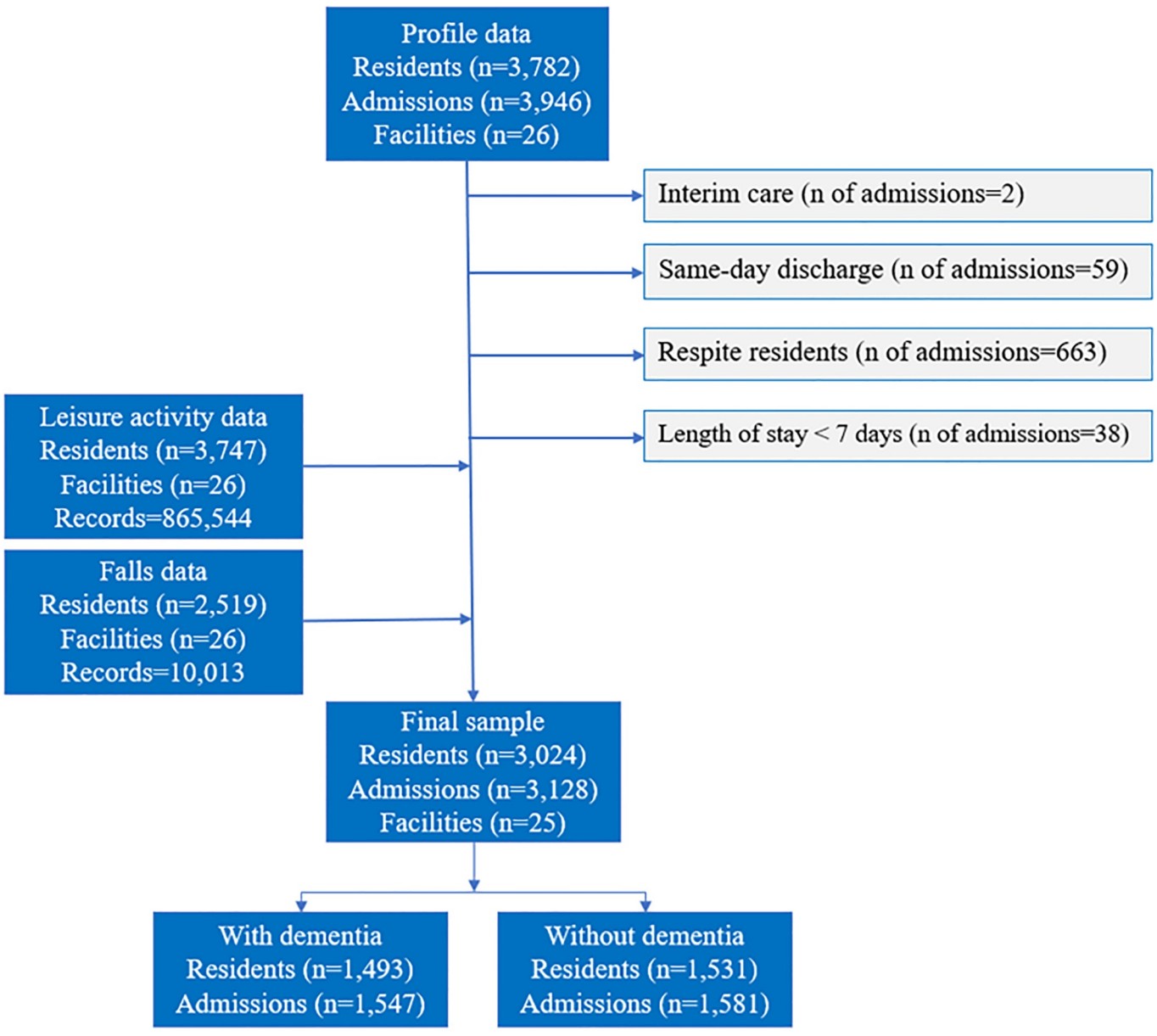

**Fig 1. Sample selection flowchart.**

admissions. Records of leisure activities of permanent residents during any previous respite admissions were excluded.

We first described the average rate of leisure activities and falls for the whole sample and separately by dementia status. We then conducted multilevel negative binomial regression to investigate the relationship between leisure participation and falls, controlling for age, gender, fall history and comorbidity status at time of admission (i.e., dementia, depression, cognitive impairment, anxiety, cerebrovascular accident, diabetes mellitus, visual impairment, delirium, wounds and Parkinson's disease), which have been demonstrated factors affecting fall risk [15, 18, 25, 38–40]. We used the three quartile values of the leisure participation rate to classify leisure participation level into four categories: low (from the lowest level of leisure participation

**Table 1. Categorization of leisure activities.**

| Non-exercise activities | Exercise activities |
| --- | --- |
| —Floor games, table games, card games, bingo, happy hour, trivia/quiz, armchair travel, singing, celebration, cooking, watching movie, listen to music, reading book/ poetry, knit and knatter, birthday bash, arts and crafts, Tovertafel, playing domino, playing piano<br>—Club activities (e.g., men's/movie/music/book club), coffee/video/virtual/face chat, visiting friends/families, zoom chat/meeting, catching up, join webinars/forum<br>—Bible study/reading/reflections, church service, devotion, hymn singing, pastoral visit, praise/pray, catholic activities, Easter celebration, holy communion service | —Exercise, walking, dancing, Tai-chi, yoga, ball sports, bowling, Zumba, cycling, chair-chi, tennis, golf, small group of Olympics, aerobic activities, wrestling, gardening |

rate to the first quartile, i.e., 0–1.59 activities per week, which equates to a range of no activities to an activity every 4.43 days, for all sample); middle (from the first quartile to the second quartile, i.e., 1.59–3.01 activities per week, or an activity every 2.33–4.43 days, for all sample); middle-high (from the second quartile to the third quartile, i.e., 3.01–5.32 activities per week, or an activity every 1.32–2.33 days, for all sample) and high (from the third quartile to the highest level of leisure participation rate, i.e., 5.32–14.31 activities per week, or an activity in less than 1.32 days, for all sample). Each category contains approximately equally one fourth of the whole sample. We used multilevel negative binary regression with random intercept for facilities as the dependent variable in our analysis (i.e., fall incidence rate) is over-dispersed The overdispersion of fall incidence rate is examined using the overdispersion test (t = 4.67, p-value<0.001) in Stata via the command 'overdisp'. and that participants were organized in two levels (i.e., individual level and facility level). The two-level modelling structure indicates that fall incidents could be vulnerable to the clustering effect, implying a possible correlation between incidents happening within the same facility. We used incidence rate ratio (IRR) to compare differences in fall incidence rate at different levels of leisure participation. Tests were two-tailed and statistical significance was set as $p$-value<0.05. Stata version 17.0 (StataCorp LP, College Station, Texas 77845 USA) was used.

## Results

### Participant characteristics

Of the total 3,024 resident sample, approximately two-thirds (n = 2,020) were female and the median age was 88 (interquartile range [IQR] 82–94) years. The median length of stay was 420 (IQR 168–606) days. Approximately half of residents had a fall history before admission (47.85%), lived with dementia (49.37%) or had depression (43.85%), while approximately one quarter of the residents had anxiety (32.04%), cerebrovascular accident (24.44%) or diabetes mellitus (27.25%) (Table 2).

### Rate of leisure participation and falls

Table 3 presents the rate of leisure participation for all residents by type of leisure activity and participants' dementia status. The rate of participation in all leisure activities was 549.97 activities per 1,000 resident days, indicating residents overall engaged in one leisure activity every two days. The majority of the leisure activities were non-exercise (474.70 activities per 1,000 resident days), compared to exercise activities (75.25 activities per 1,000 resident days). The rate of participation in all leisure activities was significantly higher among residents with

**Table 2. Sample characteristics.**

| | All sample (n = 3,024) | Dementia status | | |
| --- | --- | --- | --- | --- |
| | | With dementia (n = 1,493) | Without dementia (n = 1,531) | *p*-value |
| Female, n (%) | 2,020 (66.80) | 997 (66.78) | 1,023 (66.82) | 0.981 |
| Age, median (IQR) | 88 (82–94) | 88 (83–93) | 89 (82–94) | 0.185 |
| Age category, n (%) | | | | |
| 65–79 | 503 (16.63) | 221 (14.80) | 282 (18.42) | <0.001 |
| 80–84 | 474 (15.67) | 257 (17.21) | 217 (14.17) | |
| 85–89 | 695 (22.98) | 362 (24.25) | 333 (21.75) | |
| 90–94 | 737 (24.37) | 388 (25.99) | 349 (22.80) | |
| ≥ 95 | 615 (20.34) | 265 (17.75) | 350 (22.86) | |
| Length of stay (days), median (IQR) | 420 (168–606) | 459 (207–606) | 372 (140–606) | <0.001 |
| Having fall history, n (%) | 1,447 (47.85) | 747 (50.03) | 700 (45.72) | 0.018 |
| Health status, n (%) | | | | |
| Dementia | 1,493 (49.37) | - | - | - |
| Depression | 1,326 (43.85) | 681 (45.61) | 645 (42.13) | 0.054 |
| Cognitive impairment | 1,202 (39.75) | 673 (45.08) | 529 (34.55) | <0.001 |
| Anxiety | 969 (32.04) | 460 (30.81) | 509 (33.25) | 0.151 |
| Cerebrovascular accident | 739 (24.44) | 345 (23.11) | 394 (25.73) | 0.093 |
| Diabetes mellitus | 824 (27.25) | 357 (23.91) | 467 (30.50) | <0.001 |
| Visual impairment | 504 (16.67) | 231 (15.47) | 273 (17.83) | 0.082 |
| Delirium | 382 (12.63) | 233 (15.61) | 149 (9.73) | <0.001 |
| Parkinson's disease | 169 (5.59) | 77 (5.16) | 92 (6.01) | 0.308 |
| Wounds | 198 (6.55) | 82 (5.49) | 116 (7.58) | 0.021 |

Note: 1. p-value is based on the results of chi-square test (for nominal variables) or Wilcoxon rank-sum test (for interval variables).

dementia compared to those without dementia (575.96 *vs* 524.63 activities per 1,000 resident days, $p = 0.002$). The rates of participation in exercise and non-exercise activities were also significantly higher among residents with dementia compared to those without dementia (494.69 *vs* 455.21, $p = 0.005$; 81.27 *vs* 69.42, $p<0.001$).

Table 3 presents the incidence rate of falls. For the whole sample, the incidence rate of all falls was at 12.60 per 1,000 resident days, while the incidence rate of injurious falls was at 4.59 per 1,000 resident days. Incidence rates of both these fall groups were significantly higher among residents with dementia compared to those without dementia (14.43 *vs* 10.82 per

**Table 3. Average incidence rate of leisure activities and falls, per 1,000 resident days.**

| | All sample (n = 3,024) | Dementia status | | |
| --- | --- | --- | --- | --- |
| | | With dementia (n = 1,493) | Without dementia (n = 1,531) | *p*-value |
| **Leisure activities** | | | | |
| All leisure activities | 549.97 | 575.96 | 524.63 | 0.002 |
| *Non-exercise activities* | 474.70 | 494.69 | 455.21 | 0.005 |
| *Exercise activities* | 75.27 | 81.27 | 69.42 | <0.001 |
| **Falls** | | | | |
| All falls | 12.60 | 14.43 | 10.82 | <0.001 |
| Injurious falls | 4.59 | 4.96 | 4.23 | <0.001 |

Note: 1. p-value is based on the results of Wilcoxon rank-sum test.

1,000 resident days for all falls, *p*<0.001; 4.96 *vs* 4.23 per 1,000 resident days for injurious falls, *p*<0.001).

## Relationship between leisure participation and falls

Fig 2 presents the results of multilevel negative binomial regression to examine the relationship between leisure participation and fall risk. For all residents, the rate of participating in all leisure activities was significantly inversely associated with the incidence rate of falls after controlling for covariates; thus as participation in activities increased, fall incidence decreased. For example, compared to those with a low level of leisure participation (i.e., <25th percentile, or no activities to an activity every 4.43 days), residents in high leisure participation group (i.e., ≥75th percentile, or an activity in less than 2.33 days) were 26% less likely to experience all falls (IRR at 0.74 [95% CI = 0.60, 0.91]) and were 40% less likely to experience injurious falls (IRR at 0.60 [95% CI = 0.47, 0.76]). An inverse association was observed for participation in both non-exercise and exercise leisure activities. For example, residents with a high level of participation in non-exercise activities were 27% less likely to experience all falls compared to those in the low participation group (IRR at 0.73 [95% CI = 0.59, 0.90]); and residents with a high level of participation in exercise activities were 25% less likely to experience all falls compared to those in the low exercise group (IRR at 0.75 [95% CI = 0.61, 0.92]).

Fig 2 also demonstrates that the inverse association between leisure participation and fall risk was particularly strong among residents without dementia. Specifically, among the cohort

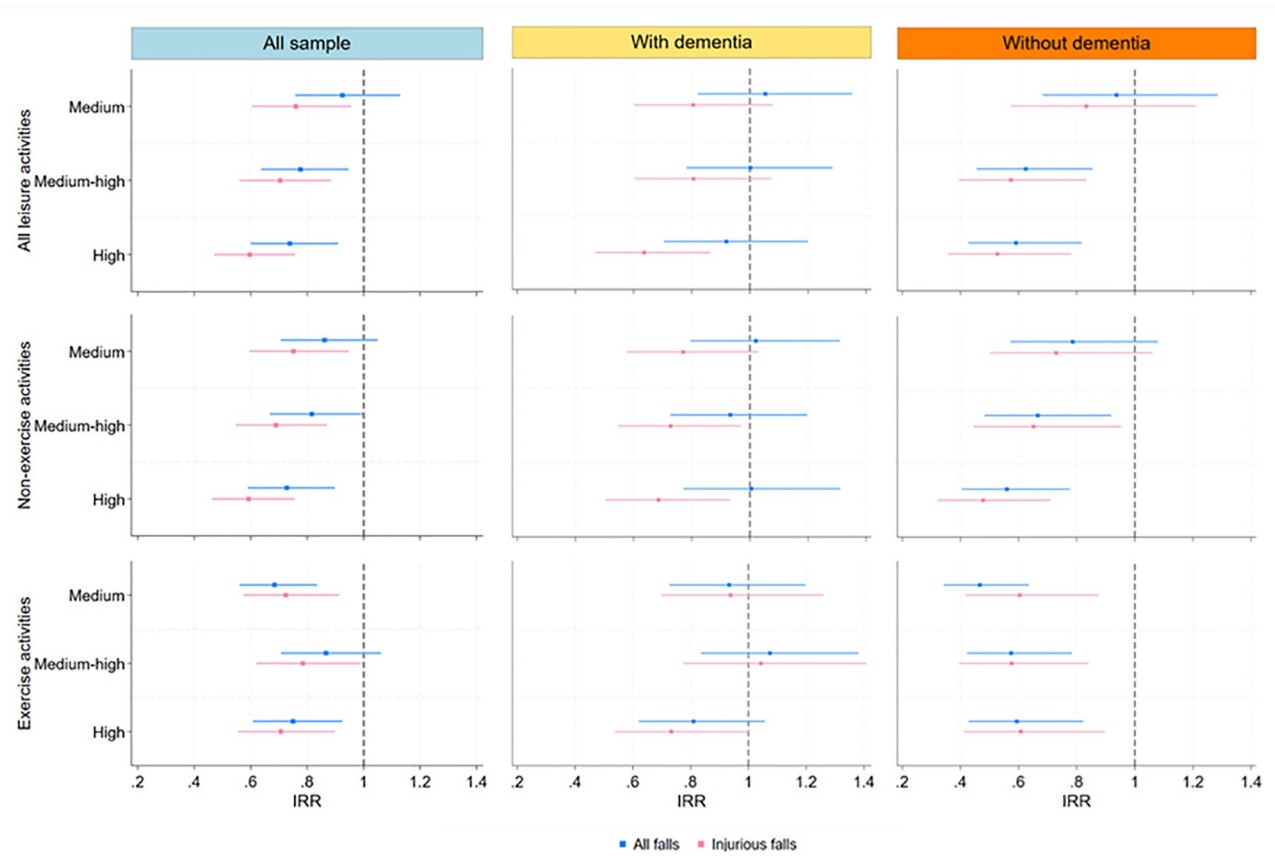

**Fig 2. The relationship between leisure activities and falls incidence, all sample and by dementia status.**

without dementia, residents with medium-high and high leisure participation, respectively, experienced a lower risk of all falls (IRR at 0.62 [95% CI = 0.46, 0.85] and 0.59 [95% CI = 0.43, 0.82], respectively) compared to those with low leisure participation. Similarly, among the cohort without dementia, residents with medium-high and high leisure participation had a lower risk of injurious falls (IRR at 0.57 [95% CI = 0.39, 0.83] and 0.53 [95% CI = 0.36, 0.78], respectively) compared to those with low participation. By contrast, leisure participation was also significantly associated with a lowered risk of all falls among the cohort with dementia; however, there were no associations between leisure participation and injurious fall outcomes for residents with dementia, regardless of the type of leisure activities.

## Discussion

Active engagement in leisure activities has been widely considered as an essential component of successful aging [12, 14]. In Australian RACFs, leisure activities are increasingly recognized as an important aspect of care provision, with many facilities employing staff (e.g., activity co-ordinators) to provide these activities supported by other care staff, such as carers, nurses and allied health professionals [41]. However, understanding of the relationship between leisure participation and fall outcomes in older adults has been inconclusive [20, 21]. This study has demonstrated that leisure participation was common among RACF residents (approximately one leisure activity every two resident days) and that, importantly, engagement in leisure activities, both exercise and non-exercise, was significantly associated with a lower incidence of falls. Further, the significant association between leisure activities and fall risk was stronger among the residents without dementia compared to those with dementia.

The finding of an association between leisure participation and a reduced risk of fall (both all falls and injurious falls) aligns with previous studies reporting a lowered fall incidence rate associated with participation in a physical exercise program [22, 23, 42]. However, our study provides new evidence, demonstrating that the association between leisure participation and reduced fall risk is not limited to exercise activities but also exists for non-exercise leisure activities in the RACF setting. This is inconsistent with previous studies reporting an inverse association between sedentary behaviour and risk of falls among older adults [43, 44]. One possible explanation is that the non-exercise leisure activities in our study encompass various activities, some of which are not entirely sedentary (e.g., cooking, social visits, and attending church services). Furthermore, the previous studies were conducted in non-RACF settings, rendering their findings less directly applicable due to differences in the characteristics of the populations. Although we found an association, the mechanism of impact is unclear. It is likely that the inverse association between non-exercise leisure participation and fall risk can be explained by non-exercise leisure activities generally only requiring moderate exercise of the musculoskeletal system. Moderate involvement of skeletal muscle movement contributes to enhanced muscle strength, balance, postural stability, gait and mobility [20], and hence, can lead to a lower fall risk.

The finding of the inverse association between leisure participation and fall risk also corresponds to the theory of activity (aging) that proposes a positive relationship between being socially and physically active and better wellbeing and life satisfaction in old age [45]. Thus, the reduced fall risk associated with leisure participation might also be explained by the improved overall wellbeing of older adults as a result of staying active and maintaining social interaction. Previous studies have demonstrated participation in physical and recreational activities might contribute to a delay in the onset and progression of frailty and chronic pain [46] and improved mental health [5, 6], better physical function [8–10] and longevity [7]. In particular, engagement in leisure activities, both exercise and non-exercise, can exert a positive

effect on older adults' mental, sensory and cognitive capacity [47], which assists to improve balance and coordination. Moreover, older adults with active leisure participation might also have greater knowledge, skills and confidence about managing their health [48]. Such overall positive effects of leisure participation on health outcomes could improve the wellbeing of older adults, and hence, contribute to a reduced risk of falls.

Additionally, leisure participation helps older adults develop new networks and increase autonomy, which are important for coping with changes surrounding the transition and adaption from the home environment to RACFs. Previous evidence has demonstrated that older adults actively engaged in leisure activities adjust better psychologically and socially to their new life in RACFs [49]. This might help older adults to overcome some fall risk factors, such as unfamiliarity with the new physical environment in RACFs (e.g., pathways and stairs) and anxiety, loneliness and depression caused by the relocation to RACFs, and hence, reduce their fall risk.

However, it is important to note that the relationship between leisure participation and fall outcomes might be two-way, which means that fall outcomes also affect residents' intention to engage in leisure activities. Specifically, residents who experienced falls may be less able or likely to participate in leisure activities partly due to a fear of falls. Such endogeneity problem (i.e., a reciprocal causation relationship between two factors) might result in biased and inconsistent estimates of parameters, and hence, might also confound the effect of leisure participation.

We found that residents with dementia exhibit a higher rate of leisure participation than their dementia-free peers and that the association between leisure participation and fall risk was stronger among residents without dementia. The higher rate of leisure participation among residents with dementia might be because residents with dementia are generally encouraged by RACF staff to join in lifestyle and leisure activities to help them maintain a good quality of life [50]. Some leisure activities, such as listening to music and therapeutic group singing, are often specifically designed for residents with dementia to address adverse their mood and social behaviours, particularly during the sundowning period when residents with dementia might experience increased disorientation, wandering, hyperactivity, aggressive behaviour and anxiety [51]. Regarding the weak effect of leisure activities in reducing falls among residents with dementia, it might be a result of the behavioural and language disturbance accompanied with dementia. Specifically, dementia can cause problems of mobility, balance and muscle weakness [52]. which might offset the effect of leisure participation in improving mobility ability and muscle strength. In addition, dementia generally leads to difficulties in communication [52], which might limit the level of social interaction and sensory and cognitive stimulation during leisure participation, hence restraining effect of leisure participation on falls. However, this study is unable to verify such assumptions.

## Implications for future studies

While this study identifies an inverse relationship between leisure participation and falls, we do not know the mechanism through which leisure participation, particularly non-exercise leisure participation, affects fall outcomes. Future research using study designs to examine the causal relationship between leisure activities and falls, such as a randomized controlled trial or longitudinal cohort studies are required. Ideally, residents' characteristics, such as religious beliefs, social past and role in their families, need to be considered in future studies to create a consumer-centered leisure and lifestyle program. It would also be of interest to discern the effect of different types of non-exercise leisure activities on falls. Different types of non-exercise leisure activities, such as those for socializing (e.g., visiting friends) and those for

entertainment (e.g., playing card games) or religious and spiritual activities (e.g., church service), might affect fall outcomes in different ways and directions. Such knowledge will be important for RACFs to develop tailored leisure activities for effective fall prevention and improved outcomes.

## Strengths and limitations

We conducted a retrospective longitudinal cohort study on the relationship between leisure participation and falls in RACFs using routinely collected data from 25 Australian RACFs. The strengths of this study lie in its large sample size (i.e., 3,024 participants) and its comprehensive measure of leisure activities (both exercise and non-exercise) and fall outcomes (all falls and injurious falls). Additionally, we stratified our analysis by dementia status, a dimension that has not been investigated in the existing literature.

This study has several limitations. First, the data used were collected from one aged care service provider in Sydney, New South Wales, Australia, and therefore, may not be representative of other RACF populations. Second, based on a retrospective cohort study design, this study is unable to control for all risk factors of falls in the analysis since the datasets used were not initially constructed for the research purpose of falls so not all pertinent risk factors of falls were recorded. For example, due to data limitations, this study has not included factors at the organizational level, such as the location of facility, the number of staff and the availability of registered nurses, which can affect falls. Third, the information on chronic conditions was collected on the date of admission, which might not accurately reflect changes in health conditions throughout the study period. Fourth, given that the aged care provider did not record the ending time of each leisure participation, we are unable to incorporate the length of time spent in each leisure activity in our analysis; this might miss the differences in the extent/level of participation among different residents. Fifth, only association and not causation can be inferred from our results. As discussed, the direction of the association between leisure participation and fall outcomes cannot be discerned from our results based on a retrospective cohort design. Sixth, dementia is a progressive disease and has different stages (e.g., early, moderate and severe); therefore, the level of leisure participation might vary among the stages of dementia within the group of residents with dementia. However, due to data limitations, we are unable to control the different stages of dementia in the analysis. Seventh, residents' decision to engage in exercise or non-exercise activities might be affected by their physical functional status, which could vary over the period of admission. Given a lack of longitudinal data, we were unable to control for any differences in the physical functional status in relation to engagement in exercise and non-exercise activities in our analysis. Eighth, by using secondary data, we are unable to eliminate the potential possibility of inconsistencies in data collection among the 25 RACFs during the five-year study period. However, given that all the 25 RACFs are administered by one single care provider throughout the study period and that data were collected electronically through the same information system of the aged care provider on a day-to-day basis, such inconsistencies can be minimized.

## Conclusions

This study provides a comprehensive investigation into the relationship between leisure participation and falls in RACF setting using a longitudinal cohort dataset in 25 Australian RACFs. The results demonstrate that a high level of engagement in leisure activities, both exercise and non-exercise, is associated with a reduced fall risk compared with a low level of leisure engagement, though such linkage is stronger among residents without dementia. Further studies are recommended to better discern the causal effect of leisure participation on fall outcomes

among the RACF residents to help better develop ongoing fall prevention and management programs.

## Acknowledgments

We thank the partners and collaborators of this project including Anglicare, Northern Sydney Local Health District, Sydney North Primary Health Network, the Deeble Institute for Health Policy Research, and the Australian Aged Care Quality and Safety Commission.

## Author Contributions

**Conceptualization:** Guogui Huang, Nasir Wabe.

**Data curation:** Guogui Huang, Nasir Wabe.

**Formal analysis:** Guogui Huang.

**Funding acquisition:** Nasir Wabe, Magdalena Z. Raban, Karla Seaman, Johanna I. Westbrook.

**Project administration:** Nasir Wabe, Johanna I. Westbrook.

**Writing – original draft:** Guogui Huang.

**Writing – review & editing:** Guogui Huang, Nasir Wabe, Magdalena Z. Raban, S. Sandun Malpriya Silva, Karla Seaman, Amy D. Nguyen, Isabelle Meulenbroeks, Johanna I. Westbrook.

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
