## [Decision Letter · Decision Letter 0]

31 Jan 2024

PONE-D-23-42031The relationship between participation in leisure activities and fall incidence in residential aged carePLOS ONE

Dear Dr. Huang,

Thank you for submitting your manuscript to PLOS ONE. After careful consideration, we feel that it has merit but does not fully meet PLOS ONE’s publication criteria as it currently stands. Therefore, we invite you to submit a revised version of the manuscript that addresses the points raised during the review process.

The manuscript does not reach enough level for acceptance in the journal.See the reviewers’ suggestions carefully and respond to them appropriately.

We look forward to receiving your revised manuscript.

Kind regards,

Masaki Mogi

Academic Editor

PLOS ONE

Journal Requirements:

"This work was supported by the Australian National Health and Medical Research Council (NHMRC) Partnership Project Grant (1170898). The funding organization did not have any influence on the study design, data collection, analysis and interpretation as well as the preparation, review, or approval of the manuscript for publication."

3. In the online submission form, you indicated that [The data used in this study are available from the aged care provider (Anglicare), but restrictions apply to the availability of these data, which were used under license for the current study, and so are not publicly available. Data are however available from the authors (Guogui Huang, guogui.huang@mq.edu.au) upon reasonable request and with permission of Anglicare.]. 

5. Please ensure that you refer to Figure 2 in your text as, if accepted, production will need this reference to link the reader to the figure.

6. Please include a copy of Table 4 which you refer to in your text on page 13.

Reviewers' comments:

Reviewer's Responses to Questions

**Comments to the Author**

1. Is the manuscript technically sound, and do the data support the conclusions?

Reviewer #1: Yes

Reviewer #2: Yes

2. Has the statistical analysis been performed appropriately and rigorously? 

Reviewer #1: Yes

Reviewer #2: I Don't Know

3. Have the authors made all data underlying the findings in their manuscript fully available?

Reviewer #1: No

Reviewer #2: Yes

4. Is the manuscript presented in an intelligible fashion and written in standard English?

Reviewer #1: Yes

Reviewer #2: Yes

5. Review Comments to the Author

Reviewer #1: Thank you for the opportunity to review this manuscript, titled ‘The relationship between participation in leisure activities and fall incidence in residential aged care’. The study found an association between quantity of leisure activities (exercise and non-exercise), and reduced fall risk. The study found this association was stronger in residents living with dementia, compared to those not living with dementia. The manuscript was well written and framed. I understand my comments below may be limited by data availability.

1. Methods: have the falls outcomes in this dataset been validated at all?

2. I note that the leisure activities data contains “residents’ participation level” - can this be used to say anything about dose-response? Firstly, for example, activities that have some sort of meaning to the participant vs something to pass the time. Were there enough ‘attended but refused to participate’ to investigate around this? Why would residents attend but not participate - do residents self-select activities? Who drives leisure activities? Do RACFs have targets or KPIs around activities engagement? Line 381 – raises also the question of ‘quality’ of activity or level of participation

Secondly, do you have data to investigate whether there was a difference in time spent on leisure activities? E.g. between 15 mins vs 5 hours of non-exercise leisure?

Thirdly, can you look into variety at all? E.g. going to Church 3x/week vs doing 3 different activities/week? E.g. one strong social network vs a potentially wider, more varied one?

3. Classification – ‘medium high’ vs ‘high’ – doesn’t take into account time spend on leisure activity, i.e. 1 activity for 5 hrs vs 3 activities for 20 mins each? More time taken in activities = more time around others who can help watch them/prevent falls during that time (i.e. less time at risk)

It might also be helpful to put these classifications in terms that are easier to grasp e.g. Low is no activities to an activity every 4.42 days (this leads to the question: would it be better to have classifications that make more practical sense? You could end up with uneven sized groups but x activities every (whole) day is easier to understand and interpret?

4. There's some mention of loneliness in your ms, but not much? This emerging field/concept seems quite relevant (and related to my comment #2 in terms of quality and time of activities).

5. Please clarify numbers in Fig 1: leisure activity data for n=3747, the missing n=35 didn’t do any activities or were they missing data? Same for falls data n=2519, missing n=1263. Also N=762 excluded (for interim care, discharged on same date etc), n=3782-762=3020 which is > final sample 3024?

6. Table 4 (mentioned in line 282) doesn’t exist? I think this refers to Fig 2?

7. Would be good to see the results refered to in Fig 2 note 2 - perhaps in supp material?

Reviewer #2: I appreciate the effort put into your manuscript on the association between leisure activities and fall incidence. After careful consideration, I find that the paper requires a major revision for the following reasons:

Clarity in Methodological Justification:

In the Introduction section, the manuscript emphasizes the significance of considering the impacts of both dementia and "depression" on the association between leisure activities and fall incidence. However, the rationale for performing statistical analyses specifically on the basis of dementia is vaguely explained. It is crucial to either include the analysis related to depression or provide a more detailed and explicit logic in the Introduction section, focusing on dementia. This will enhance the overall coherence and comprehensibility of the study design.

Lack of Data and Limitation Acknowledgment:

The manuscript categorizes leisure activities into "exercise activity" and "non-exercise activity" without presenting data on participants' physical function. This classification might be dependent on physical function, and the relevant data should be shown and included in the analyses in the study.

6. PLOS authors have the option to publish the peer review history of their article (what does this mean?). If published, this will include your full peer review and any attached files.

Reviewer #1: No

Reviewer #2: No

---

## [Author Response · Author response to Decision Letter 0]

6 Mar 2024

We have carefully considered the constructive comments provided by the editor and the two reviewers and have substantially revised the manuscript accordingly. All the major amendments are highlighted in the revised manuscript. 

Our detailed responses are outlined in the document ‘Response to Reviewers’, which has been uploaded with the resubmission of the manuscript.

---

## [Decision Letter · Decision Letter 1]

8 Apr 2024

The relationship between participation in leisure activities and incidence of falls in residential aged care

PONE-D-23-42031R1

Dear Dr. Huang,

We’re pleased to inform you that your manuscript has been judged scientifically suitable for publication and will be formally accepted for publication once it meets all outstanding technical requirements.

Kind regards,

Masaki Mogi

Academic Editor

PLOS ONE

Additional Editor Comments (optional):

The authors well responsed to the Reviewers' suggestions.  No further comment.

Reviewers' comments:

Reviewer's Responses to Questions

**Comments to the Author**

1. If the authors have adequately addressed your comments raised in a previous round of review and you feel that this manuscript is now acceptable for publication, you may indicate that here to bypass the “Comments to the Author” section, enter your conflict of interest statement in the “Confidential to Editor” section, and submit your "Accept" recommendation.

Reviewer #1: All comments have been addressed

Reviewer #2: All comments have been addressed

2. Is the manuscript technically sound, and do the data support the conclusions?

Reviewer #1: Yes

Reviewer #2: Yes

3. Has the statistical analysis been performed appropriately and rigorously? 

Reviewer #1: Yes

Reviewer #2: I Don't Know

4. Have the authors made all data underlying the findings in their manuscript fully available?

Reviewer #1: No

Reviewer #2: Yes

5. Is the manuscript presented in an intelligible fashion and written in standard English?

Reviewer #1: Yes

Reviewer #2: Yes

6. Review Comments to the Author

Reviewer #1: Thank you for thoughtful revisions. I note the data availability has been restricted by the ethics approval for this project, but data may be available upon reasonable request.

Reviewer #2: Authors have responded to all of my comments accurately and revised the manuscript carefully based on the response.

7. PLOS authors have the option to publish the peer review history of their article (what does this mean?). If published, this will include your full peer review and any attached files.

Reviewer #1: **Yes: **Esa Chen

Reviewer #2: No
